# Relationship between Dietary Patterns and Subjectively Measured Physical Activity in Japanese Individuals 85 Years and Older: A Cross-Sectional Study

**DOI:** 10.3390/nu14142924

**Published:** 2022-07-17

**Authors:** Tao Yu, Yuko Oguma, Keiko Asakura, Michiyo Takayama, Yukiko Abe, Yasumichi Arai

**Affiliations:** 1Graduate School of Health Management, Keio University, 4411 Endo, Fujisawa City 252-0883, Japan; yuto.kojima@keio.jp; 2Sports Medicine Research Center, Keio University, 4-1-1 Hiyoshi, Kohoku-ku, Yokohama City 223-8251, Japan; 3Department of Environmental and Occupational Health, School of Medicine, Toho University, 5-21-16 Omorinishi, Ota-ku, Tokyo 143-8540, Japan; jzf01334@nifty.ne.jp; 4Center for Preventive Medicine, Keio University, 35 Shinanomachi, Shinjuku-ku, Tokyo 160-8582, Japan; michiyo@keio.jp; 5Center for Supercentenarian Medical Research, Keio University, 35 Shinanomachi, Shinjuku-ku, Tokyo 160-8582, Japan; yukiko_abe@keio.jp (Y.A.); yasumich@keio.jp (Y.A.); 6Faculty of Nursing and Medical Care, Keio University, 4411 Endo, Fujisawa City 252-0883, Japan

**Keywords:** aging population, healthy behaviors, dietary patterns, physical activity, healthy life expectancy

## Abstract

High-quality diets and regular physical activity (PA) are considered healthy behaviors (HBs). HBs are associated with many health outcomes and are expected to improve quality of life. Although implementing HBs is important, the relationship between dietary patterns (DPs) and PA has not been well investigated, especially among those aged ≥ 85. This study used data from the Tokyo Oldest Old survey on Total Health study to examine the relationship between DPs and PA in a cross-sectional study. The dietary survey used the brief self-administered diet history questionnaire to estimate the intake of 58 foods. After energy adjustment, principal component analysis was performed to identify major DPs. A validated questionnaire was used to evaluate PA, and linear regression analysis was used to investigate the association between DPs and PA, considering confounders. A total of 519 participants were included. Three major DPs (‘Various plant foods’, ‘Fish and mushrooms’, ‘Cooked rice and miso soup’) were identified. ‘Various plant foods’ was similar to DPs previously named ‘Healthy’ or ‘Prudent’, and its trend was positively associated with higher PA. This study observed the implementation of HBs even among those aged ≥ 85, suggesting that a trend toward a healthier diet is associated with higher PA.

## 1. Introduction

Diet and physical activity are closely related to energy intake and consumption behaviors and are summarized as energy balance-related behaviors (EBRBs) [1]. Consuming a high-quality diet and engaging in high physical activity (PA) are considered healthy behaviors (HBs) and have been reported to be associated with health-related outcomes (such as quality of life, cognitive function, development of disease, and functional disability) in a variety of populations [2,3,4,5,6,7,8,9].

Focusing on diet rather than specific nutrients is informative because the daily diet is not limited to a single nutrient. Previous studies have not considered the interactions of multiple nutrients in a diet in the body [10]. Many studies have used factor analysis or principal component analysis (PCA) to identify major dietary patterns (DPs) and examined their association with health outcomes, such as physical performance [11]. DPs vary and are influenced by region, age, and other factors. For example, in Japan, more than 285 DPs have been reported to date, and these can be summarized into six major DPs, such as healthy patterns or Western patterns, based on their similarity [12]. There is no consistent definition of a healthy diet, but previous studies have shown that healthy DPs often include eating more plant foods (such as vegetables, mushrooms, and fruits) and foods nutritionally rich in vitamins and minerals. Trends in DPs considered healthy in each country are positively associated with physical performance [13]. PA is essential in maintaining physical and mental health [14]. It has also been identified as a protective factor against various diseases [15]. Some estimates suggest that if the world’s population was more active, 4–5 million deaths could be avoided annually. The World Health Organization launched guidelines on physical activity and sedentary behavior in 2020 [16]. Metabolic equivalents (METs) are often used to measure PA intensity. METs*h/week is a unit of PA volume (intensity × duration) based on the number of times more energy is expended in that exercise when the resting oxygen uptake of 3.5 mL/kg/min is set as 1 [17]. More PA has been associated with a lower risk of mortality [18].

With the global population living longer, the number of older adults aged ≥ 85 is expected to increase. An important task is to increase healthy life expectancy, which allows for independent living, and the implementation of HBs is considered crucial for this purpose. Public health interventions targeting PA and dietary habits are superior in cost-effectiveness [19]. In the past, either diet or PA was examined with health outcomes; however, few studies have been associated with the two. In addition, previous studies have focused on adults and younger older adults (≥65), and considerably fewer studies have been conducted on adults aged ≥ 85 [20,21]. Because behavioral patterns related to energy balance play an essential role in influencing the health of older adults, understanding their association can guide interventional research on older adults and is a necessary component of successful aging. Therefore, this study aimed to identify DPs and their relationship with PA in populations aged ≥85.

## 2. Methods

### 2.1. Study Population

This cross-sectional study used baseline data from the Tokyo Oldest Old Survey on Total Health (TOOTH) study conducted in 2008–2009. A total of 3320 potential participants were randomly selected from the basic city registry of adults aged ≥ 85 living in central Tokyo (within a 6 km radius of the Keio University Hospital). Of these, 542 agreed to participate in the medical and dental surveys at the university hospital in addition to the general and dietary surveys and were included in this study. Excluding those with missing data in the dietary survey, the total number of participants for analysis was 519 (Figure 1). For more details, please refer to the research protocol [20]. The TOOTH Study was approved by the ethics committee of Keio University School of Medicine (ID: 20070047) and registered in the UNIN-Clinical Trial Registry (ID: UMIN000001842). Participation in the TOOTH Study and the survey were conducted in accordance with the Declaration of Helsinki, and written informed consent was acquired from participants interested in the survey after a thorough explanation. Only the participants who provided their consent were included in the study and were able to withdraw their consent at any time upon request.

### 2.2. Diet Survey and Identification of Dietary Patterns

The brief self-administered diet history questionnaire (BDHQ), which has been validated in the older population aged ≥ 80, was used in this study [21]. The questionnaire estimated energy and nutrient intake based on the type, amount, and frequency of foods consumed in a typical meal during the past month. The BDHQ (and other self-administered questionnaires) were mailed to participants who completed the interview sessions and were willing to participate. The BDHQ was conducted by the participant. For participants who had difficulty, a family member acquired responses verbally and completed the questionnaire. Participants could access the study staff by telephone when they experienced difficulties or obscurity in answering the questionnaires. The completed BDHQs were checked by experienced staff (such as an experienced geriatrician) for data quality, consistency, and completeness.

There are two main types of DP identification methods: a priori DP, which is based on established hypotheses, and a posteriori DP, which is data-dependent [10]. An example of a priori DP is the Mediterranean diet score, which has been reported to be associated with physical performance. Still, it is unclear whether it reflects the daily diet in Japan due to the innate characteristics of the Mediterranean diet and regional differences [22]. Other studies have assessed adherence to the Japanese version of the Dietary Balance Guide and examined its association with mortality; however, these studies were sensitive to the accuracy of dietary surveys [23]. A posteriori DP is a technique that uses PCA, factor analysis, or cluster analysis for DP, but the naming of patterns is left to the investigators of each study. However, because it is useful to characterize the population, PCA was used to identify the DPs in this study.

### 2.3. Physical Activity

PA was estimated using the modified Zutphen Physical Activity Questionnaire (PAQ), which was validated in this age group [24]. PAQ is based on the walking speed (fast, normal, and slow), walking time, and activity intensity for each time and type of exercise and is multiplied by the number of hours per day and number of times per week to calculate the amount of PA (METs*h/week). This study used three outcomes: walking, exercise (such as calisthenics and resistance training), and PA index (PAI), the sum of these activities, in METs*h/week.

### 2.4. Assessment of Covariates

Participants were queried about age, sex, activities of daily living (ADL, evaluated by the Basel Index), education years, economic status, employment status, living conditions, and smoking habits. Medical history (cardiac disease (angina pectoris, myocardial infarction, coronary heart disease, atrial fibrillation, heart failure), chronic diseases (diabetes, hypertension, dyslipidemia, kidney disease), and cancer) was ascertained. Cognitive function was assessed by a professional psychologist in a private room using the Mini-Mental State Examination (MMSE), and body mass index (BMI) was calculated by measuring weight and height (weight (kg)/height (m)^2^).

### 2.5. Statistical Analysis

Quantitative variables (such as age or BMI) are shown as medians (25th–75th percentile), and categorical variables (such as sex or living alone) are shown as the number and percentage (%) of persons in each category. Group comparisons were analyzed by the Mann–Whitney U test and Chi-square test (or Fisher’s exact test). Before identifying DPs, we classified the 58 foods into 33 foods/food groups, which were estimated based on a previous study of Japanese participants aged 70–90 [25]. These were energy-adjusted using the density method, and PCA was performed. Based on eigenvalues and scree plots, up to the third principal component that could be interpreted as DP was examined, and a principal component score was calculated. The principal component scores were continuous variables ranging from −1 to 1. Higher scores indicate greater adherence to DP. The independent variable was the principal component scores for each DP, the outcome variable was PA, and their relationships were examined in the linear regression model. Sex, age, BMI, ADL, and MMSE were added to Model 1. Education years, living status, economic status, smoking habits, and medical history were added to Model 2. Statistical analysis was performed using SPSS version 26.0 (IBM Japan, Tokyo, Japan), and statistical significance was defined as *p* < 0.05.

## 3. Results

A total of 519 participants were included for analyses, and their characteristics are shown in Table 1. Participants were included in the analysis, with a median age of 87.3 years and a BMI of 21.4. The MMSE and ADL scores were 27 and 100, respectively. Of the total respondents, 33.9% lived alone, and 72.8% reported that their economic status was good or very good (see Table 2 for DPs details). PCA identified three major DPs. The 1st DP was characterized by consuming plant foods such as green and dark yellow vegetables or other vegetables and was named ‘Various plant foods’. The 2nd DP was characterized by the intake of fish and seafood as well as mushrooms and was called ‘Fish and mushrooms’. The 3rd DP was characterized by the consumption of cooked rice and miso soup (a traditional Japanese soup), so this pattern was named ‘Cooked rice and miso soup’. Three DPs accounted for 12.4, 7.2, and 5.6% of the variance, respectively (Table 2).

The high-trend group in the 1st DP showed significantly higher values in walking, exercise, and PAI. The 2nd DP showed differences in exercise, and the 3rd DP showed no significant differences between the two groups. The 1st DP showed higher protein, fat, fiber, and most micronutrients (vitamins and minerals) in the high-trend group (the group that scored higher than the median on the PCA), whereas carbohydrate intake was low. The 2nd DP showed nearly the same trend as the 1st DP. The 3rd DP showed no significant differences in protein and carbohydrates; however, fat was significantly lower in the high-trend group. The micronutrients showed different trends for different items (Appendix A).

The relationship between the DPs and PA is presented in Table 3. Although the 2nd and 3rd DPs were not associated, the 1st DP was associated with PAI (1.41, 0.33–2.48 (B, 95% CI)). Furthermore, the results suggested an association with exercise rather than walking (0.64, 0.02–1.25).

## 4. Discussion

In this study, DPs were identified, and their relationship to PA was examined in adults aged ≥ 85. DPs were identified as ‘Various plant foods’, ‘Fish and mushrooms’, and ‘Cooked rice and miso soup’. The 1st DP, ‘Various plant foods’, was positively associated with PAI. Furthermore, the results suggested an association with exercise rather than with walking. This is the first study to examine the relationship between DPs and PA in this age group to the best of our knowledge. The 1st, 2nd, and 3rd DPs were similar to the DPs named ‘Healthy/Prudent’, ‘Japanese’, and ‘Traditional Japanese’, respectively [12]. ‘Healthy’ DPs have been reported and promoted for their importance with PA and the association between implementation of these and various health-related outcomes.

Among Asian participants with an average age of 69.9 years, the presence/absence of healthy dietary behaviors derived from six nutrient groups; (1) miscellaneous, (2) milk and dairy products, (3) vegetables, (4) fruits, (5) soybean/fish/meat/egg, and (6) nuts, seeds, oil, and fat, was examined and compared with subjectively and objectively measured PA [26]. According to subjective measures, the ‘Healthy diet’ group had significantly longer total leisure time (LT), PA, and LT walking than the ‘No healthy diet’ group. According to objective measures, light PA was significantly longer among the ‘Healthy diet’ group. Although not significant, the ‘Healthy diet’ group engaged in more objectively measured PA. Those results suggest an association with exercise, not walking, so there may be differences between that study and ours based on age and survey methodology. However, in both studies, the dietary behaviors trend was positively associated with PA, with consistent results. In addition, a positive association between fruit and vegetable intake and LTPA or PA levels was reported in an Australian population (3644 participants, 48% men, mean age 60.2 years) and a Netherlands population (2466 participants, 56% men, mean age 62 years), suggesting that fruit and vegetable intake and increased PA may have a significant impact on the health of the older population [27,28]. Although fewer studies focus on older adults aged ≥ 85, one study in Croatia (Zagreb City) used the Elderly Dietary Index Score to assess the intake of ten foods (meat, fish, fruits, vegetables, grains, legumes, olive oil, alcohol, bread, and dairy products), evaluating the association with PA [29]. Those results indicated that an ‘optimal’ intake of meat, seafood, grains, fruits, legumes, and bread was associated with ‘sufficient’ PA. However, PA is not necessarily associated with a healthy diet, as a positive correlation may exist between the consumption of snacks and PA [30]. Several physical or socioeconomic factors (such as BMI, income, marital status, and residential status) determine EBRBs or HBs. In a population aged ≥ 85, education played the most important role among many factors, such as sex, BMI, disease history, cognitive function, and marital status, with differences in DPs [31]. In addition, some individuals face challenges, including geographic information of residence, access to foods and locations to engage in PA, ethnic-specific cultural values and dietary habits, and commercial influences [30,31,32,33]. Nevertheless, the trend toward a healthier diet has been positively associated with PA in different countries and ages. Moreover, this study identified this association in a previously unobserved population aged ≥85.

The frailty cycle is a brief way to understand the relationship between DPs and PA in older populations [34]. Frailty is a reversible status of significance in older populations and is characterized by muscle loss and activity decline. As a result of decreased daily activity and lower energy needs, the person has reduced appetite and eats less. Eventually, it leads to phenotypic outcomes such as weight loss and muscle mass loss, which in turn cause exhaustion, muscle weakness, and walking speed reduction. This becomes a vicious cycle, leading to a further decrease in activities, leading to further declines in eating. The positive association between DP and PA may be a state of energy need and appetite due to higher PA. This relationship needs to be clarified in future studies. Various reports on the relationship between frailty and diet have asserted that the higher the intake of vegetables and fruits, the lower the risk of frailty [35,36]. Plant foods such as vegetables and fruits contain anti-inflammatory components, are associated with inflammatory markers, and positively impact sarcopenia and frailty [37,38]. In addition, this food group is rich in micronutrients, including vitamin C, which have antioxidant properties. They protect against unnecessary muscle degradation caused by oxidative stress in the body [39]. While these hypotheses are undeniable, PA also plays an essential role in inflammatory markers and oxidative stress; therefore, populations that consume more plant-based foods are also more physically active, which may have a positive impact [40]. Hence, it is necessary to consider not only diet or PA, but both, such as EBRBs or HBs, in a comprehensive way in examining health-related outcomes. Errors in filling out the dietary survey may exist. The developers of the BDHQ have proposed that estimated energy intake above 4000 Kcal or below 600 Kcal may be overestimated or underestimated, but none of the participants in this study displayed such numbers. As a sensitivity analysis, we excluded participants with possible cognitive decline (MMSE ≤ 21, *n* = 50 (9.8%)) and re-identified dietary patterns and confirmed the association between DPs and PA; however, the results did not change. These findings suggest associations between dietary habits and PA, which are critical topics in EBRBs or HBs, and can fill a research gap that has existed until recently. In addition, they provide insight for designers and practitioners of behavior change for health promotion, allowing for more effective intervention trials and social implementation.

### Study Strengths and Limitations

This study has several strengths. First, it focused on DPs, which may be more easily applied to actual behavior. Second, this study used a linear regression model to examine several important confounders, including demographic and socioeconomic variables, lifestyle habits, and medical information. Third, studies with more than 500 participants aged ≥ 85 are rare worldwide. However, this study has several limitations. First, this was a cross-sectional study, and causal relationships cannot be established. Second, the survey was limited to those who live in central Tokyo and could visit the study site. Thus, the generalizability of the results warrants further study. Therefore, other investigations, including longitudinal studies, are needed. Third, the diet and PA surveys were subjective assessments and may have been subject to recall bias and over or underestimation. Because of the seasonality of dietary intake, methods to consider seasonality in future studies are required. Objective evaluations using accelerometers will be needed to assess PA objectively.

## 5. Conclusions

This study observed the implementation of HBs even among those aged ≥85, suggesting that a trend toward a healthier, more food-diverse DP is associated with higher PA. Future studies should examine the outcomes of the implementation of HBs. It is necessary to consider not only diet or PA but both, such as EBRBs or HBs, comprehensively in examining health-related outcomes. A questionnaire that comprehensively surveys diet and PA may be required for practical applications.

## Figures and Tables

**Figure 1 nutrients-14-02924-f001:**
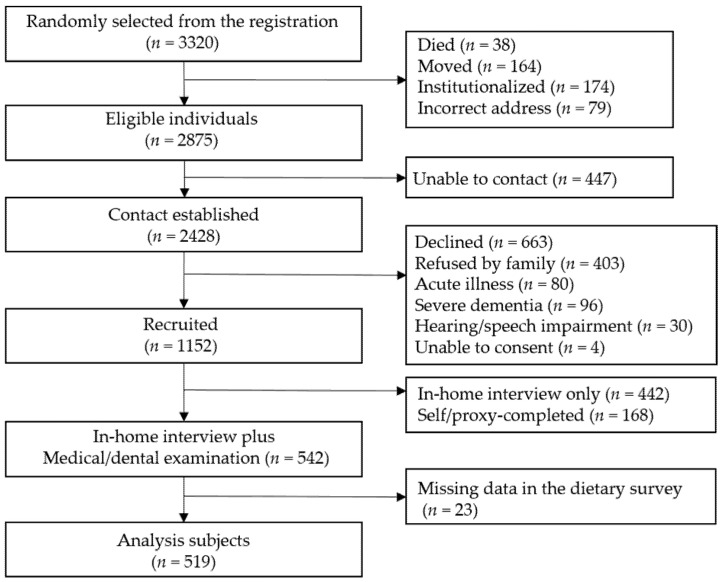
Recruitment of participants.

**Table 1 nutrients-14-02924-t001:** Participant characteristics.

		DP1 Various Plant Foods			DP2 Fish and Mushrooms			DP3 Cooked Rice and Miso Soup		
	All Participants (*n* = 519)	Low Trend Group (*n* = 260)	High Trend Group (*n* = 259)	*p*-Value	Low Trend Group (*n* = 260)	High Trend Group (*n* = 259)	*p*-Value	Low Trend Group (*n* = 259)	High Trend Group (*n* = 260)	*p*-Value
Sex										
Men	219 (42.2)	125 (48.1)	94 (36.3)	0.01	128 (49.2)	91 (35.1)	<0.001	104 (40.2)	115 (44.2)	0.37
Women	300 (57.8)	135 (51.9)	165 (63.7)		132 (50.8)	168 (64.9)		155 (59.9)	145 (55.8)	
Age	87.3 (86.2–88.8)	87.3 (86.1–88.7)	87.4 (86.3–88.9)	0.73	87.3 (86.2–88.6)	87.4 (86.4–89.0)	0.23	87.3 (86.1–88.7)	87.3 (86.4–88.8)	0.42
Body Mass Index ^(*n* = 517)^	21.4 (19.4–23.6)	21.3 (19.4–23.5)	21.4 (19.3–23.7)	0.97	21.5 (19.4–23.7)	21.2 (19.2–23.3)	0.44	21.6 (19.4–23.7)	21.1 (19.3–23.3)	0.27
Mini-Mental State Examination ^(*n* = 510)^	27 (25–29)	27 (24–29)	27 (25–29)	0.26	28 (25–29)	27 (24–29)	0.03	27 (25–29)	27 (24–29)	0.62
Activities of daily living ^(*n* = 511)^	100 (95–100)	100 (95–100)	100 (100–100)	0.02	100 (95–100)	100 (95–100)	0.66	100 (95–100)	100 (100–100)	0.21
Year of education ^(*n* = 497)^	11 (9–13)	11 (8–13)	11 (9–13)	0.44	11 (10–14)	11 (8–13)	<0.01	11 (10–14)	11 (8–13)	<0.01
Living alone ^(*n* = 505)^	171 (33.9)	91 (36.0)	80 (31.7)	0.35	83 (32.7)	88 (35.1)	0.57	90 (36.2)	81 (31.5)	0.26
Smoking habit ^(*n* = 502)^										
Smoker	35 (7.0)	17 (6.8)	18 (7.1)	0.24	26 (10.3)	9 (3.6)	<0.001	18 (7.2)	17 (6.7)	0.91
Ex-smoker	161 (32.1)	89 (35.6)	72 (28.6)		91 (36.1)	70 (28.0)		78 (31.2)	83 (32.9)	
Non-smoker	306 (61.0)	144 (57.6)	162 (64.3)		135 (53.6)	171 (68.4)		154 (61.6)	152 (60.3)	
Economic status ^(*n* = 499)^										
Very good/Good	363 (72.8)	184 (73.9)	179 (71.6)	0.22	181 (72.1)	182 (73.4)	0.52	181 (72.4)	182 (73.1)	0.87
Neither	78 (15.6)	42 (16.9)	36 (14.4)		37 (14.7)	41 (16.5)		41 (16.4)	37 (14.9)	
bad/Very bad	58 (11.6)	23 (9.2)	35 (14.0)		33 (13.1)	25 (10.1)		28 (11.2)	30 (12.0)	
Working ^(*n* = 498)^	94 (18.9)	37 (15.2)	57 (22.4)	0.04	51 (20.3)	43 (17.4)	0.43	45 (18.1)	49 (19.7)	0.73
No disease history ^(*n* = 485)^	101 (20.8)	58 (23.6)	43 (18.0)	0.22	55 (22.2)	46 (19.4)	0.64	49 (20.9)	52 (20.8)	0.34
PAI, METs*h/week	7.0 (2.0–14.7)	6.3 (1.5–12.0)	8.8 (3.3–17.0)	<0.001	7.0 (1.7–14.5)	7.1 (2.7–14.7)	0.15	7.0 (1.9–14.0)	7.0 (2.3–15.1)	0.79
Walking, METs*h/week	4.2 (1.5–10.5)	3.5 (1.0–9.8)	4.9 (2.0–11.2)	<0.01	4.2 (1.4–10.5)	4.5 (1.8–10.5)	0.56	4.2 (1.5–9.8)	4.7 (1.5–10.5)	0.58
Exercise, METs*h/week	0.0 (0.0–4.0)	0.0 (0.0–2.9)	0.0 (0.0–4.5)	<0.01	0.0 (0.0–3.0)	0.0 (0.0–4.1)	0.01	0.0 (0.0–4.1)	0.0 (0.0–3.5)	0.97

The effective number of participants is shown next to the item; values are shown as median (25th–75th percentile) or number (%). The *p*-value is a test of the difference between the high- and low-trend groups for each dietary pattern (DP); low- or high-trend group means scored lower or higher than the median on principal component analysis. PAI: physical activity index (sum of walking and exercise). Body Mass Index is calculated by weight (kg)/(height)^2^ (m^2^), Barthel Index evaluates activities of daily living, and disease history includes heart disease, kidney disease, cancer, hypertension, diabetes, and dyslipidemia. See Table 2 for DP details.

**Table 2 nutrients-14-02924-t002:** Identification of dietary patterns.

Food Item	DP1 Various Plant Foods	DP2 Fish and Mushrooms	DP3 Cooked Rice and Miso Soup
Cooked rice	−0.43		0.68
Noodles			
Bread		−0.54	
Miso soup	−0.24		0.58
High-fat milk			
Low-fat milk			
Red meats			
Chicken		0.25	
Processed meats			−0.23
Fish		0.57	
Shellfish		0.32	
Seafood		0.27	
Egg			
Potatoes	0.23	0.35	
Soy products	0.25	0.29	0.35
Green and dark yellow vegetables	0.70	0.24	
Other vegetables	0.80		
Pickled vegetables	0.44		0.29
Salad vegetables	0.67		
Mushrooms	0.46	0.48	
Seaweeds	0.31	0.38	
Fruit	0.21		
Confectioneries	−0.28		−0.52
Ice cream			−0.24
Sugar	0.31	−0.57	
Fats and oils	0.75		0.03
Alcoholic beverages			
Green tea			0.41
Black and Oolong tea			−0.35
Coffee	0.21	−0.59	
Soft drinks			−0.30
Fruit and vegetable juice			−0.32
Seasonings	0.56		0.41
Total	4.09	2.37	1.85
Initial eigenvalues % of Variance	12.41	7.17	5.60
Cumulative %	12.41	19.58	25.17

Numbers indicate the loading each food group or food accounts for, and items with an absolute value < 0.20 are left blank. DP, dietary pattern.

**Table 3 nutrients-14-02924-t003:** The relationship between dietary patterns and subjectively measured physical activity.

	PAI, METs*h/week						Walking, METs*h/week						Exercise, METs*h/week					
	Model 1			Model 2			Model 1			Model 2			Model 1			Model 2		
	B	95% CI	*p*-Value	B	95% CI	*p*-Value	B	95% CI	*p*-Value	B	95% CI	*p*-Value	B	95% CI	*p*-Value	B	95% CI	*p*-Value
DP1 Various plant foods (*n* = 435)	1.17	0.08–2.25	0.04	1.41	0.33–2.48	0.01	0.59	−0.21–1.39	0.15	0.78	−0.03–1.57	0.06	0.57	−0.04–1.19	0.07	0.64	0.02–1.25	0.04
DP2 Fish and mushrooms (*n* = 435)	0.54	−0.54–1.61	0.33	0.49	−0.59–1.57	0.37	0.17	−0.63–0.96	0.68	0.19	−0.61–0.99	0.64	0.37	−0.24–0.98	0.24	0.30	−0.32–0.92	0.34
DP3 Cooked rice and miso soup (*n* = 435)	0.09	−0.99–1.17	0.87	−0.01	−1.08–1.07	0.99	0.21	−0.59–1.00	0.61	0.14	−0.66–0.93	0.74	−0.12	−0.73–0.49	0.71	−0.14	−0.76–0.48	0.66

Model 1 was adjusted for sex, age, BMI, ADL, and MMSE. Model 2 was adjusted for years of education, living status and economic status, smoking habit, and medical history, in addition to the variables in Model 1. B: Partial regression coefficient; CI: confidence interval; PAI: physical activity index (sum of walking and exercise).

## Data Availability

The data presented in this study are available upon request and approval by the ethics committee of Keio University School of Medicine.

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
