# Peer review of "Relationship between Dietary Patterns and Subjectively Measured Physical Activity in Japanese Individuals 85 Years and Older: A Cross-Sectional Study"

_nutrients, 2022, doi:10.3390/nu14142924_

Round 1

Reviewer 1 Report

These are my remarks and suggestions:

Female study participants are not found in Table 1 and should be introduced as well;

Please provide some information on Ethical Considerations;

Page 3, Line 121. „55 foods into 33 foods/food groups” Provide the food groups used in the dietary pattern analysis (Food Groups/Dietary Items);

Pages 9-10. Please provide the Strength and Limitations of the current study as a separate subchapter;

Overall, more discussion and comparisons with the literature on the results are needed.

In the Conclusions section, add more information on the practical application of the results obtained during the research and specify this study's novelty.

Author Response

Dear Reviewer:

We wish to re-submit the manuscript titled “Relationship Between Dietary Patterns and Subjectively Measured Physical Activity in Japanese Individuals 85 Years and Older: a cross-sectional study.” The manuscript ID is nutrients-1801043.

We thank you for your thoughtful suggestions and insights. The manuscript has benefited from these insightful suggestions. I look forward to working with you to move this manuscript closer to publication in the Nutrients.

The manuscript has been rechecked, and the necessary changes have been made by your suggestions. Revisions to the main manuscript are indicated in yellow highlight. The responses to all comments have been prepared and attached in our response letter.

Sincerely,

Yuko Oguma

Graduate school of Health Management, Keio University

4-1-1 Hiyoshi Kouhoku-ku, Yokohama City, Kanagawa 223-8251, Japan,

Email: [email protected],

Tel: 045-566-1090

Fax: 045-566-1090

Reviewer 2 Report

First of all, I would like to thank you for sending me this manuscript for review and congratulate the authors for their research in this much-needed area. However, the year of data collection is considered to be very old, more than a decade ago, which means a decrease in the current interest or generability.  Even so, below, I indicate a series of comments for the improvement of the manuscript.

It is recommended to use the Strobe Statement.

Indicate the research design in the title or abstract.

Review punctuation and capitalization after period.

The criteria for inclusion and exclusion of participants are not indicated.

Sample and power size should be included.

The conditions under which the subjects were measured should be further described.

Indicate who performed the measurements and number of investigators, information given to participants, etc.

Ethics committee approval is not included.

No registration in Clinical Trial or similar is included.

Include potential sources of bias.

Author Response

(The authors gave the same response as above.)

Round 2

Reviewer 2 Report

Thank you for the review and correct.